# Developing the Urban Thermal Environment Management and Planning (UTEMP) System to Support Urban Planning and Design [†]

**Dongwoo Lee [1] and Kyushik Oh [2],***

[1]   Research Institute of Spatial Planning & Policy, Hanyang University, Seoul 04763, Korea;
     estevan97@hanyang.ac.kr
[2]   Department of Urban Planning and Engineering, Hanyang University, Seoul 04763, Korea
*   Correspondence: ksoh@hanyang.ac.kr; Tel.: +82-2-2220-0336
†   This article is a revised and extended version of the conference paper presented at the 2nd International
     Conference on Sustainability, Human Geography and Environment, Kraków, Poland,
     28 November–2 December 2018 (ICSHGE 2018).

**Abstract:** Mathematical Climate Simulation Modeling (MCSM) has the advantage of not only investigating the urban heat island phenomenon but also of identifying the effects of thermal environment improvement plans in detail. As a result, MCSM has been applied worldwide as a scientific tool to analyze urban thermal environment problems. However, the meteorological models developed thus far have been insufficient in terms of their direct application to the urban planning and design fields due to the preprocessing task for modeling operations and the excessive time required. By combining meteorological modeling and Geographic Information System (GIS) analysis methods, this study developed the Urban Thermal Environment Management and Planning (UTEMP) system that is user-friendly and can be applied to urban planning and design. Furthermore, the usefulness of UTEMP was investigated in this study by application to areas where the heat island phenomenon occurs frequently: Seoul, Korea. The accuracy of the UTEMP system was verified by comparing its results to the Automatic Weather Systems (AWSs) data. Urban spatial change scenarios were prepared and air temperature variations according to such changes were compared. Subsequently, the urban spatial change scenarios were distinguished by four cases, including the existing condition (before the development), applications of the thermal environment measures to the existing condition, allowable future urban development (the maximum development density under the urban planning regulations), and application of the thermal environment measures to allowable future development. The UTEMP system demonstrated an accuracy of adj. $R^2$ 0.952 and a ±0.91 Root Mean Square Error (RMSE). By applying the UTEMP system to urban spatial change scenarios, the average air temperature of 0.35 °C and maximum air temperature of 1.27 °C were found to rise when the maximum development density was achieved. Meanwhile, the air temperature reduction effect of rooftop greening was identified by an average of 0.12 °C with a maximum of 0.45 °C. Thus, the development of UTEMPS can be utilized as an effective tool to analyze the impacts of urban spatial changes and for planning and design. As a result, the UTEMP system will allow for more efficient and practical establishment of measures to improve the urban thermal environment.

**Keywords:** urban climate; mathematical climate simulation modeling; GIS; urban planning and design

## 1. Introduction

The urban thermal environment is increasingly worsening as climate change, urbanization, and human activities increase [1]. These thermal environmental problems are the cause of a variety of

urban problems, including decline in the health of citizens, increasing energy consumption, decline of ecosystem service functions, and deterioration of air quality [2,3]. In the case of Korea, if greenhouse gas is emitted as BaU (Business as Usual), it is expected that temperatures of about 4–5 °C will increase in most metropolitan cities after 100 years. Not surprisingly, various attempts have been made to improve the thermal environment through planning and design techniques, including the introduction of heat reduction infrastructures and the creation of wind corridors.

Meanwhile, it would be ideal to utilize long-term observational data throughout the entire urban area in order to investigate urban climatic characteristics and to analyze the effects of thermal environment improvement measures. However, this is practically impossible due to space constraints, installation time, and expensive operating costs [4]. Therefore, a Mathematical Climate Simulation Model (MCSM) that interprets natural phenomena as a mathematical equation using computers has been mainly applied to urban thermal environment analysis [5].

The MCSM is capable of quantitative analysis of complex urban thermal environments and has advantages in that similar iterative analyses can be conducted and verified relatively quickly at a lower cost than field observations or wind tunnel tests [6]. With recent advances in analytical techniques and computer processing speeds, the use of mathematical modeling continues to increase in urban planning and design [4,7].

On the meso-scale level, energy balance models such as WRF (Weather Research and Forecasting Model) and MM5 (Mesoscale Meteorological Mode version 5) have been applied to investigate overall meteorological phenomena (temperature, wind direction, wind velocity, etc.) of large urban areas, and to analyze the effects of heat environment improvement with the introduction of large parks and green areas [8–10]. In addition, a guideline map to plan and manage the thermal environment throughout urban areas has been developed that takes into account MCSM results and urban spatial characteristics [11].

On the other hand, on the micro-scale (building unit or block scale), climate variations due to the introduction of heat reduction infrastructures (vegetation and cool pavements expansion) or changes of building form and arrangement have been analyzed by applying computational fluid dynamic models (CFD) such as Envi-met, Fluent, and so on. Roth and Lim [12] investigated the usefulness and limitations of Envi-met as an urban thermal planning tool. Gromke et al. [13] identified the heat reduction effects of introducing trees and green roofs by applying the Fluent Model. Wang, Berardi, and Akbari [14] established thermal environment improvement scenarios combining albedo control, cool roofs introduction, and vegetation expansion, and their heat mitigation effects were investigated using the Envi-met model. On the pedestrian level, Ng et al. [15] analyzed the heat reduction effects of green roofs by applying the Envi-met model. In addition, Nararian, Sin, and Norford [16] simulated outdoor comfort resulting from urban spatial changes, by applying a micro scale model.

Although the utilization of MCSMs is increasing in the urban planning and design field to improve the thermal environment, in order to apply urban planning and design alternatives mainly created by Computer-Aided Design (CAD) or Geographic Information System (GIS) to the MCSM, complex pre-processing is required. In particular, most models use the model input database provided by the country where the model was developed, so important spatial characteristics are often ignored when applying climate modeling to other countries. Therefore, there are challenges for non-specialists related to climate model research to operate MCSMs. In addition, despite the progress of computer processing speed, it takes a relatively long time to identify model results [5,17], which makes it difficult to compare the alternatives that are changed frequently in the process of urban planning and design.

Furthermore, most of the climate modeling that has been developed so far is limited to either the meso-scale or the micro-scale depending on the characteristics of the model such as assumptions on meteorological phenomena. Therefore, in order to establish effective measures to improve the thermal environment, as Shuzo [18], Mirzaei [17], and Mirzaei and Haghighat [4] have emphasized that it is necessary to develop a model that integrates the meso-scale and the micro-scale.

The applicability limit of the MCSM increases the need for system development that can be usefully applied to urban planning and design to effectively improve the thermal environment. The objective of this study is to develop a SDSS (Spatial Decision Support System) to foster urban thermal environment improvement through urban planning and design, and to verify the usefulness of the developed system. Thus, this study developed the Urban Thermal Environment Management and Planning (UTEMP) system that integrates a MCSM and a GIS engine to improve the urban thermal environment through planning and design. In addition, to verify the accuracy of the MCSM that integrates meso-scale and micro-scale spatial characteristics, air temperatures were simulated on the district-scale level (between meso-scale and micro-scale), and the simulated results were compared with observed air temperatures. Finally, its usefulness for urban planning and design were identified by investigating air temperature variations according to urban spatial changes.

## 2. Materials and Methods

### 2.1. Developing the UTEMP System

The UTEMP system was established by combining a GIS engine and a thermal environment analysis engine, both of which were developed by a private Korean company. The GIS engine of the UTEMP system has technological characteristics that are internationally compatible (meeting the open geospatial consortium standard) and provides over 80 types of geo-processing tools. Meanwhile, the thermal environment analysis engine is a kind of morphological model that integrates a meso-scale and a micro-scale model. This engine is capable of performing one-hour climate analysis of large-scale urban space (about 605.24 km$^2$) in 30 m × 30 m resolution within 20 min and securing an accuracy FAC2 (Fact of two) of more than 80%.

#### 2.1.1. Main Functions of the UTEMP System

The UTEMP system consists of seven main functions and 27 sub functions. The main functions are as follows: (1) File, (2) Home, (3) Urban Planning Regulation and Climate Conditions, (4) Urban Spatial Change Simulation, (5) Thermal Environment Improvement Alternatives, (6) Thermal Environment Analysis, and (7) Thermal Environment Improvement Plan Assessment (Table 1). File and Home provide elementary GIS functions such as project creation and management, data input and removal, and geo-processing. In the case of urban planning and climate inventories, the UTEMP system provides basic information of the study area to establish thermal environment alternatives such as zoning information, urban climate zones, topography, and land cover.

**Table 1.** Main and sub functions of the Urban Thermal Environment Management and Planning (UTEMP) system.

| Main Functions | Sub Functions |
| --- | --- |
| File | Project (project creation, open project, save), data (add data, remove data, export data), conversion data, map creation |
| Home | View (pan, zoom in/out, etc.), selection (attribute/location selection), geo-processing (buffer, clip, intersect, union, etc.), measurement, view mode change |
| Urban Planning Regulation and Climate Conditions | Urban planning and management (zoning, urban facilities, land use, road, building, etc.) climate information (air temperature, wind direction, wind speed, atmospheric pressure, relative humidity, cloudiness, etc.), climate characteristics (urban climate zones), topography, statistics |
| Urban Spatial Change Simulation | Maximum development density simulation, input alternatives, applying urban development patterns, editing, statistics |
| Thermal Environment Improvement Alternatives | Creation of wind corridor (roads, buildings), heat reduction infrastructures introduction (urban parks, vegetation, cool pavements, green roofs, water spaces), establishing thermal environments scenarios |
| Thermal Environment Analysis | Climate modeling (air temperature, wind speed, wind direction), thermal comfort modeling, urban spatial statistics |
| Thermal Environment Improvement Plan Assessment | Alternatives assessment (air temperature reduction, thermal environment improvement), alternatives comparison |

The most important functions of the UTEMP system are as follows: Urban Spatial Change Simulation, Establish Thermal Environment Improvement Plan, Thermal Environment Analysis, and Thermal Environment Improvement Plan Assessment. The Urban Spatial Change Simulation function can predict whether the individual buildings of the study area can be changed to the maximum development density under urban planning regulations. In addition, if an urban development alternative plan is expected in the study area, this plan can be inputted for thermal environment analysis. The application of the typical planning and design techniques to improve the thermal environment, such as the creation of a wind corridor and the introduction of heat reduction infrastructures, is possible through the Establishing Thermal Environment Improvement Alternatives function. Finally, the Thermal Environment Analysis and Thermal Environment Improvement Plan Assessment functions enable data pre-processing for MCSM and analyze climate variations according to urban spatial changes. In other words, the UTEMP system was developed to implement both climate modeling functions and thermal environment improvement measures (Figure 1).

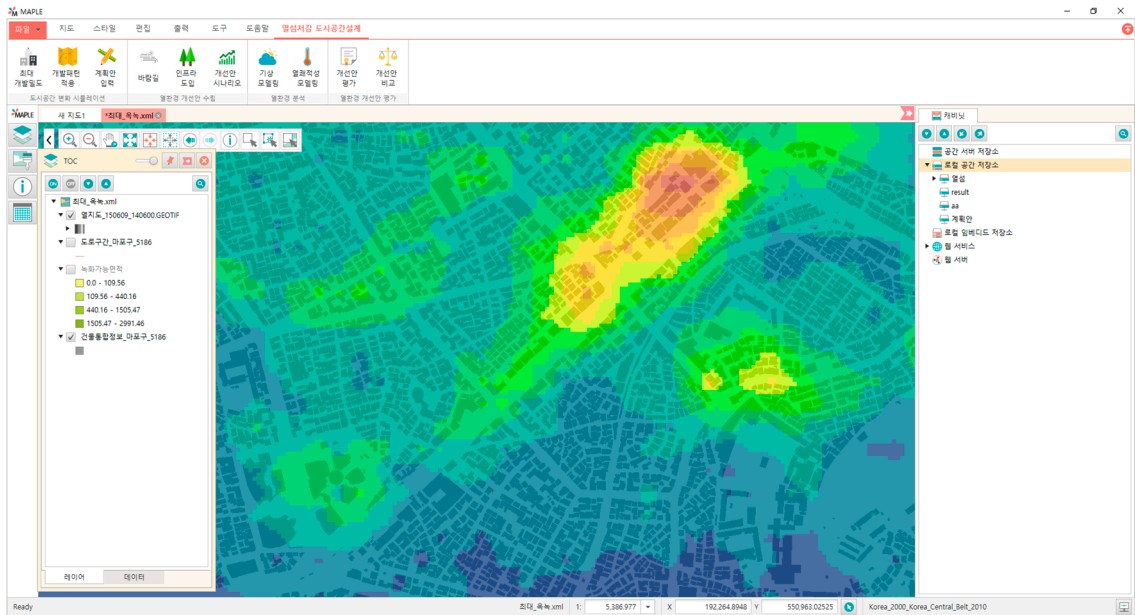

**Figure 1.** User interface of the UTEMP system.

### 2.1.2. Developing the Thermal Environment Analysis Engine

The thermal environment engine included in the UTEMP system is classified into the meso-scale climate model and the thermal dispersion model. The meso-scale climate model consists of a flat terrain model, urban model, and complex terrain model. To create a meteorological field that reflects the urban spatial structure of the modeling area, assuming that the model areas are a flat terrain, the first meteorological field was established based on the Automatic Weather System's (AWS's) observation data. Next, an urban model was created reflecting building information including location and shape, and a final meteorological field was generated reflecting the elevation information of the urban model through a complex terrain model.

On the other hand, the temperature field model was composed of a Heat Source Tag Model (HSTM) and the Lagrangian Particle Dispersion Model (LPDM). Applying Equation (1) developed by Nunez and Oke [19], the HSTM calculated the energy balance of the model area based on the meso-scale climate model.

$$Q^* + Q_F = Q_s + H + E \tag{1}$$

*Q\*: Net radiation,*
*$Q_F$ = Artificial heat*

*Qs = Storage heat flux*
*H = Sensible heat flux*
*E = Latent heat flux.*

Considering the shade effect, net radiation was calculated based on the shadow factor and the sky view factor. In order to estimate the storage heat flux, Equation (2) was applied and parameters for land covers were selected. They are presented in Table 2.

$$Q_s = \sum_{i=1}^{n} f_i a_{1i} Q^* + \sum_{i=1}^{n} (f_i a_{2i}) \frac{\partial Q^*}{\partial t} + \sum_{i=1}^{n} f_i a_{3i} \tag{2}$$

$f_i$: *Area ratio of land cover*
$Q_i$: *Net radiation*
$a_1, a_2, a_3$: *Coefficient determined by land cover type*
$t$: *hour*

**Table 2.** Parameters of land cover to estimate net radiation.

| Land Cover Type | $a_1$ | $a_2$ | $a_3$ | Sources |
|---|---|---|---|---|
| Green | 0.34 | 0.31 | −31 | Grimmond and Oke [20] |
| Building | 0.07 | 0.06 | −5 | |
| Impervious | 0.83 | 0.4 | −54.2 | Mean value of research results of Doll et al. [21] and Asaeda and Ca [22] |
| Water | 0.5 | 0.21 | −39.1 | South et al. [23] |
| Road | 0.61 | 0.41 | −27.7 | Mean value of research results of Doll et al. [21] and Asaeda and Ca [22] |

Next, to apply LPDM, the climate phenomena including air temperature, wind direction, and wind speed of the model area was predicted based on the sensible heat flux, which was calculated by HSTM. The entire model process was coded by FORTRAN (FORmular TRANslator) and was included in the UTEMP system as sub-functions.

*2.2. The Case Study*

The usefulness of the UTEMP system was verified by applying it to the Seogyo-dong area in Seoul. Although this area is not representative of the more common high-rise neighborhoods in Seoul, urban thermal environmental planning and management has been required because of Seogyo-dong's high potential for urban development. The study area is dominated by impervious surfaces with building coverage, and vegetation is very insufficient. In addition, although the Han River is adjacent to the area, ventilation is very poor due to the indiscriminate mix of both high-rise and low- rise buildings. As a result, as presented in the research results by Lee and Oh [24], this area has spatial characteristics that are likely to lead to the heat island phenomenon that is frequent among the urban climate zones in Seoul. In order to verify the UTEMP system's usefulness as a MCSM that integrates meso-scale and micro-scale characteristics, a district scale area (0.8 km$^2$) with mixed commercial and residential areas including high-and low rise buildings and frequent human activity (very active due to a nearby subway station) was investigated. In addition, to consider the boundary effect, the modeling area in the simulation was set to a much bigger area (3.5 km × 2.5 km, 8.75 km$^2$), shown as the entire air photomap in Figure 2.

To verify the usefulness of the UTEMP system to improve the urban thermal environment in urban planning and design, the case study consisted of (1) preparing and pre-processing variables to mathematical climate simulation modeling, (2) verifying the accuracy of the modeling results, and (3) identifying air temperature variations according to urban spatial changes (Figure 3).

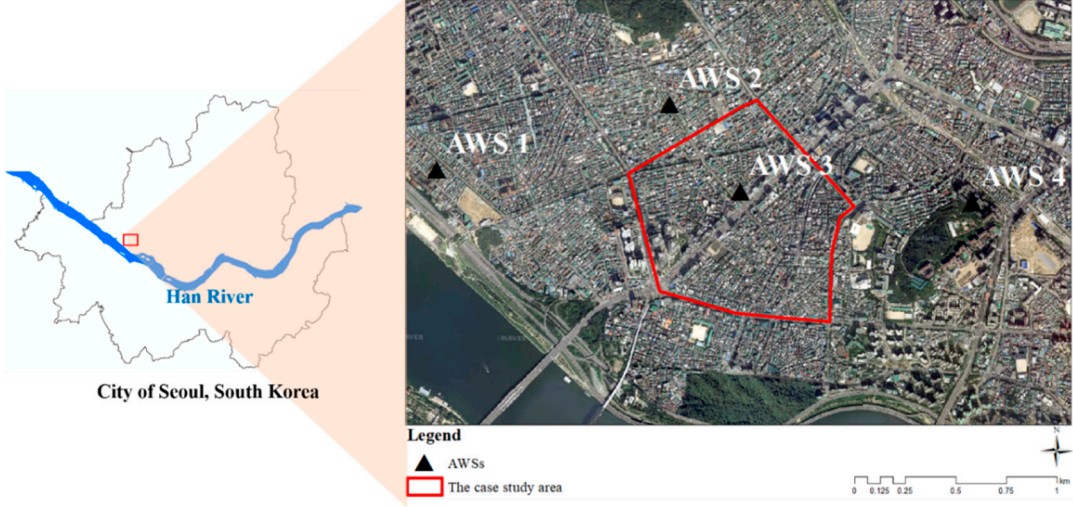

**Figure 2.** The case study area.

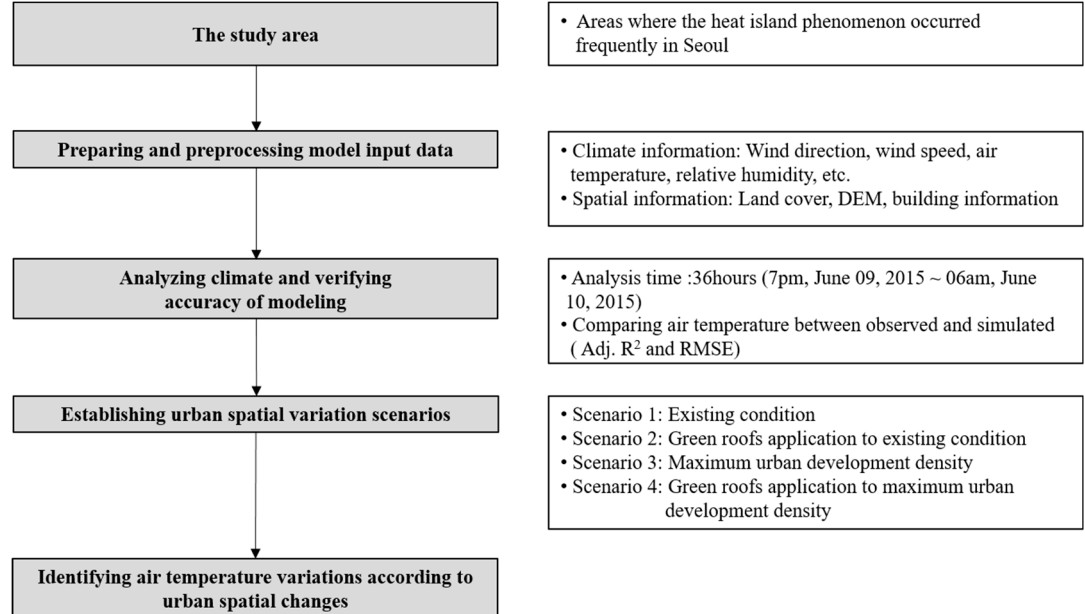

**Figure 3.** The flow of the case study. DEM = digital elevation model. RMSE = root mean square error.

### 2.2.1. Preparing and Pre-Processing Variables for MCSM

To analyze the thermal environment of urban space, inclusion of climate information, land cover, terrain, location, and shape information of buildings are essential. The national DBs (DataBases) that include climate information and spatial information provided by a Korea portal DB website were inputted as elementary DBs of the UTEMP system and were pre-processed for MCSM. Table 3 shows the elementary DBs applied to the UTEMP system. In addition, Figure 4 shows the results of converting the building information of the vector format into the ASCII (American Standard Code for Information Interchange) format for climate modeling.

**Table 3.** Input variables of the UTEMP system for urban thermal environment analysis.

| | Input Data | Usages | Spatial/Time Resolution | Sources |
|---|---|---|---|---|
| Climate information | Wind direction, wind speed, air temperature, coldness, relative humidity, precipitation | Meteorological field creation | Hour | Korea Meteorological Administration |
| Spatial information | Land cover | Heat balance estimation (Sensible heat flux, net radiation) | 1:25,000 | Ministry of Environment, Korea |
| | DEM | Wind flow analysis | 10 m × 10 m | Ministry of Land, Infrastructure and Transport, Korea |
| | Building information | Climate modeling analysis considering complex urban structures | 1:1000 | Ministry of Land, Infrastructure and Transport, Korea |

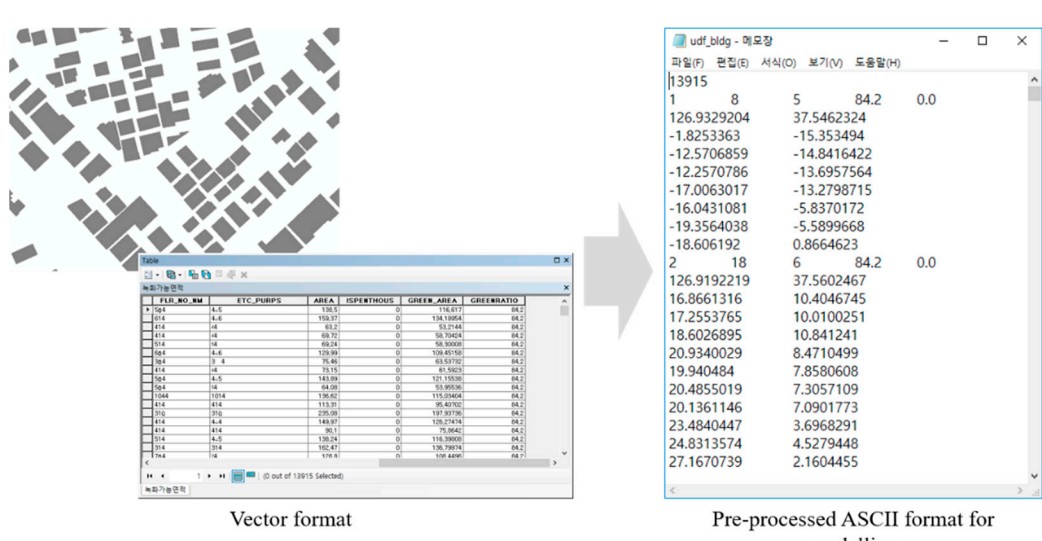

Vector format

Pre-processed ASCII format for modelling

**Figure 4.** The pre-processed results of building information (Vector to ASCII).

### 2.2.2. Accurate Verification of the Climate Model Results

In order to verify the accuracy of the MCSM, it is necessary to select a day that has minimal external effects, including clouds, precipitation, and high wind speed. In general, the summer season in the study area is June to August, and this study selected the least cloudy day with gentle wind speed during that period. As a result, it was found that 10 June 2015 was appropriate to verify model results. This study simulated climate phenomena for 36 h from 19:00 on 9 June 2015 to 06:00 on 11 June 2015. To create a meteorological field, climate information including wind speed, wind direction, air temperature, relative humidity, and cloudiness near the AWS were inputted. In addition, to identify the air temperature results in different urban spatial characteristics, data from AWSs which are located in different urban spatial characteristics were obtained. As presented in Figure 2, the AWS 1 is adjusted with rivers, and AWS 2 is located in a mixed land use area (commercial and residential) with diverse building heights. ASW 3 is located in commercial areas with mid-rise build up. AWS 4 is located among apartment residences. The simulated results were obtained with a spatial resolution of 30 m × 30 m in one-hour units. The accuracy of the modeling results was verified by confirming the adj. $R^2$ and RMSE (Root Mean Square Error) with observed data of the four AWSs in the case study area.

### 2.2.3. Application of the UTEMP System

In order to identify the applicability of the UTEMP system as a tool to improve the thermal environment, four scenarios were established and variations of air temperature due to urban spatial changes were investigated (Table 4 and Figure 5). The first scenario was an existing condition in which no urban spatial change has occurred. There are 1700 buildings in the case study area, and the average building to coverage ratio (BCR) and floor area ratio (FAR) are 41.3% and 258.8%, respectively.

**Table 4.** The urban spatial change scenarios.

| | Building (BCR [1], FAR [2]) | Heat Mitigation Measures (Applied Area) |
|---|---|---|
| Scenario 1 | Existing condition (41.3%, 258.8%) | None |
| Scenario 2 | Existing condition (41.3%, 258.8%) | Applying green roofs (99,467 m$^2$) to existing condition (scenario 1) |
| Scenario 3 | Maximum development under urban planning regulations (54.8%, 412.0%) | None |
| Scenario 4 | Maximum development under urban planning regulations (54.8%, 412.0%) | Applying green roofs (235,545 m$^2$) to maximum development (scenario 3) |

[1] BCR: Building to coverage ratio; [2] FAR: Floor area ratio.

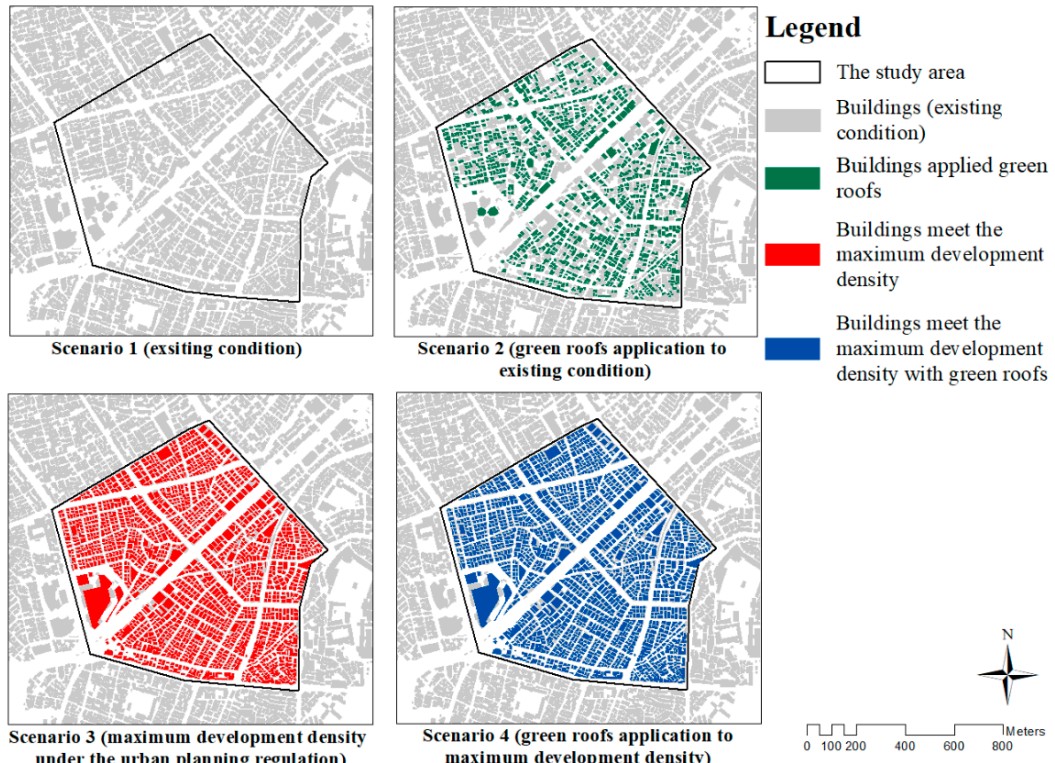

**Figure 5.** The urban spatial change scenarios.

The second scenario applied maximum green roofs to existing conditions (scenario 1) in order to improve the thermal environment. The applicability of green roofs could be determined by the roof shape and the age of the buildings [10]. Therefore, green roofs were applied only to buildings that satisfy both of the two conditions, namely the roof type was flat and the age of the building was less than 30 years. The applicable green roof areas were identified by the heat reduction infrastructure's introduction function that is included in the UTEMP system. From the results, green roofs were found to be applicable on 1096 buildings (64%) among the 1700 buildings. The total green roofs area was 99,467 m$^2$.

As a virtual urban development situation, the third scenario assumed that the individual buildings were developed with a maximum development density according to urban planning and architectural regulations. In other words, scenario 3 is a condition in which the urban thermal environment is likely to deteriorate to the maximum by urban development. This was analyzed by applying the function of a maximum development density simulation included in the UTEMP system. The simulation results revealed that the average BCR and FAR in the case study area could be increased by 54.8% (about 13.5%) and 412% (about 153.2%), respectively.

The final scenario applied green roofs to maximum development conditions (scenario 3). The reason for establishing the fourth scenario was to measure how much the thermal environment could be improved by the urban thermal environment improvement measures under the worst thermal environment conditions. Since all the buildings in scenario 3 were assumed to be newly constructed, it was assumed that the green roofs were applicable to all buildings. However, because it is impossible to introduce green rooftops in all areas of the roofs (considering the roof structures of Korean buildings), it is assumed that 70% of individual buildings could be introduced as green roofs. The results determined that green roofs could be introduced to a total of 235,454 m$^2$ in scenario 4.

## 3. Results

### 3.1. Accuracy Verification of Simulated Air Temperature

Figure 6 shows the distribution of simulated air temperatures at 00:00, 04:00, 08:00, 12:00, 16:00, and 20:00 on 10 June 2015. The average temperature of the case study area was 27.37 °C and the range was from 22.05 to 33.05 °C. As the Han River is located in the southern part of the study area, the air temperature of the southern part was relatively lower than other areas. On the other hand, it was found that the air temperature of the central parts in the case study area where it is considerably urbanized and where impervious wide roads are located, is relatively high.

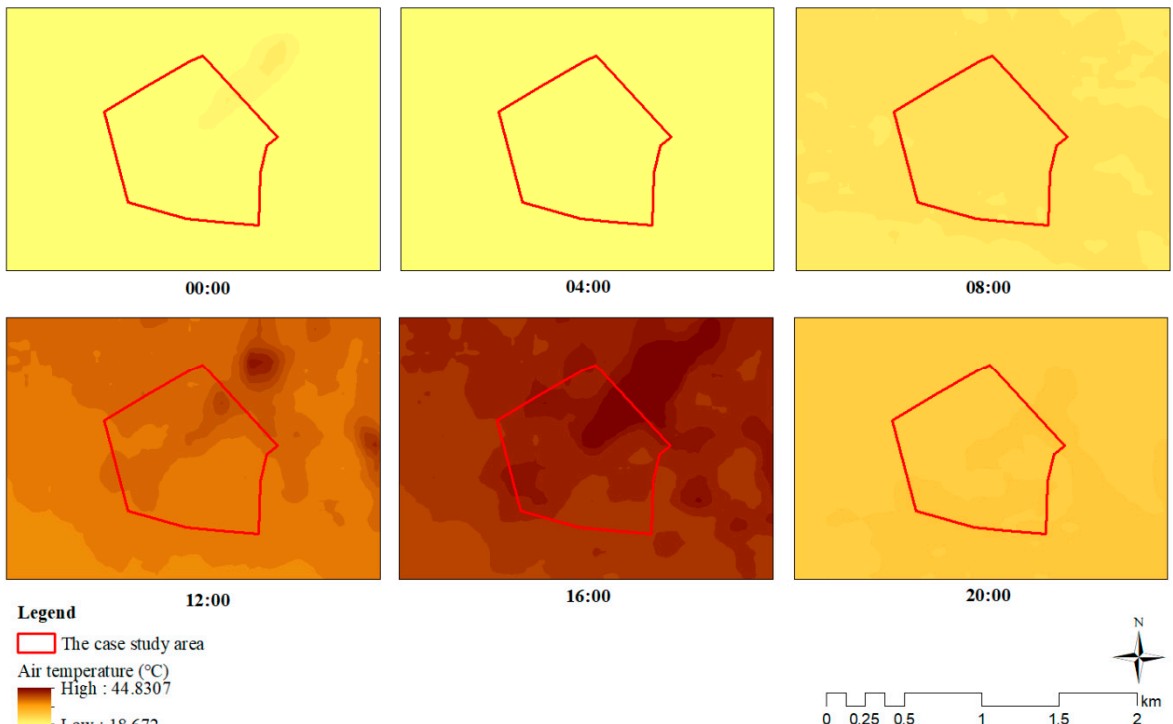

**Figure 6.** Air temperature analysis results by the UTEMP system (00:00, 04:00, 08:00, 12:00, 16:00, and 20:00, 10 June 2015).

As a result of comparing simulated air temperature and observed air temperature by the four AWSs, Adj. $R^2$ ranged from 0.923 to 0.978 with a mean of 0.952 and RMSE ranged from ±0.75 to ±1.26 °C with a mean of ±0.915 °C (Figure 7 and Table 5). The Adj. $R^2$ and average RMSE are similar to those of Roth and Lim [12] ($R^2$: 0.77~0.98, RMSE: 0.52~1.41 °C) Yang and Bou-Zeid [9] ($R^2$: 0.97, RMSE: 1.36 °C), Emmauel and Fernando [25] ($R^2$: not presented, RMSE: 1.06~2.61 °C), and He et al. [26] ($R^2$: not presented, RMSE: 1.95 °C). However, it was observed that the air temperatures at a certain time (afternoon) were somewhat higher, and such an occurrence had been discussed in many studies related to MCSM [12]. The MCSM of the UTEMP system is an initial version and accuracy verification is underway to reduce errors with the actual data.

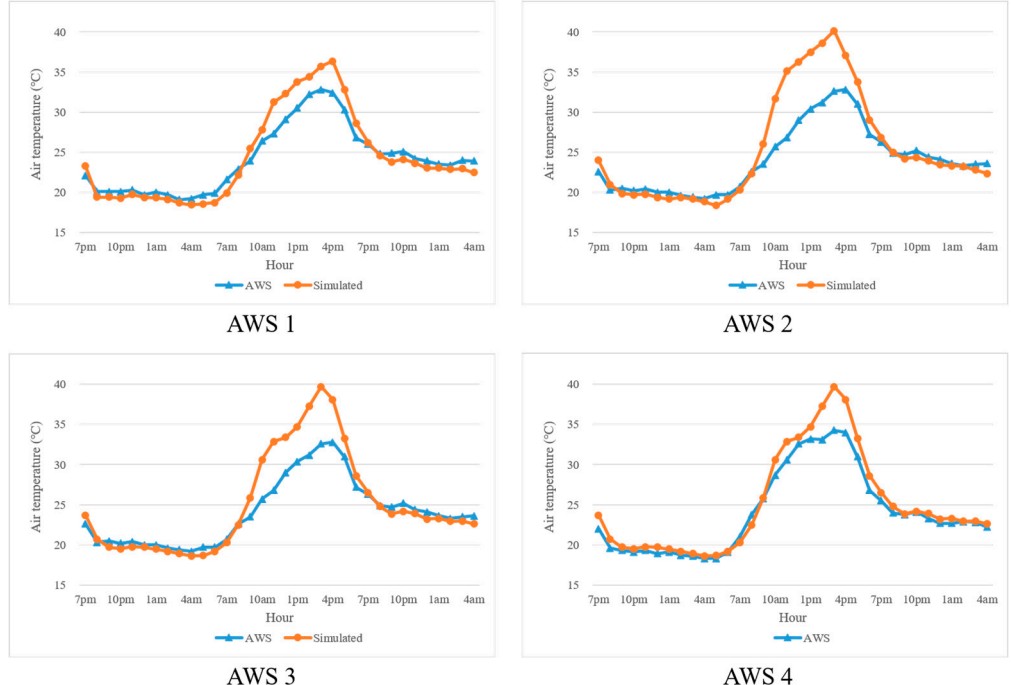

**Figure 7.** Comparison of air temperature between simulated results and observed data (Automatic Weather Systems (AWSs)).

**Table 5.** The accuracy verification results of the UTEMP system.

|  | **AWS 1** | **AWS 2** | **AWS 3** | **AWS 4** | **Mean** |
|---|---|---|---|---|---|
| Adj. $R^2$ | 0.941 | 0.966 | 0.923 | 0.978 | 0.952 |
| RMSE | ±1.00 | ±0.75 | ±1.15 | ±0.76 | ±0.91 |

*3.2. Identifying Air Temperature Variations According to Urban Spatial Changes*

From the analysis of the variations of air temperature due to urban spatial changes, the air temperature of scenario 1 (existing condition) ranged from 18.57 to 37.03 °C with an average temperature of 25.82 °C, and scenario 2 (applying green roofs) ranged from 18.62 to 36.69 °C with an average temperature of 25.77 °C. In the case of scenario 3 (maximum development), air temperature was distributed from 19.04 to 38.05 °C with an average temperature of 26.24 °C, and in scenario 4 (applying green roofs to maximum development), the air temperature was distributed from 18.48 to 37.20 °C with an average temperature of 26.11 °C (Figure 8).

When the maximum green roofs were introduced in the study area, the average temperature decreased by 0.07 °C and the maximum temperature reduction effect was the highest at 11:00 with 0.45 °C (Figure 9). Such an air temperature reduction effect due to green roofs application is similar to the other research results of Gromke et al. [13], Wang, Berardi, and Akbari [14], Kim and Oh [10],

and Ng et al. [15]. On the other hand, if the individual buildings in the study were developed to the maximum density, the difference of the average temperature was estimated to increase by about 0.35 °C. Such an air temperature difference was highest at 04:00 with 1.27 °C (Figure 10).

Additionally, when the green roofs were applied to a maximum development condition (scenario 4), the average temperature increased by 0.08 °C compared with scenario 1, and the maximum difference of average temperature was the highest at 17:00 with 0.92 °C. On the other hand, when compared with scenario 3, the average temperature was reduced by about 0.34 °C, and the effect was the maximum at 16:00 with 0.98 °C (Figure 11). In particular, the temperature reduction effects were identified mainly during nighttime (00:00 to 06:00 and 20:00 to 23:00), whereas the average nighttime temperature was lower than that of scenario 1. In the case of daytime, although the maximum temperature reduction effect was identified at 16:00, the average daytime temperature was higher than the average daytime temperature of scenarios 1 and 2. This means that the temperature-decrease effects due to the increase of the green roof area is lower than the increase effects due to the increase of the urban development density (increase of FAR and BCR) during the daytime. On the other hand, at nighttime, as the green rooftop area increases, the latent heat flux effect increases significantly [27,28], and the air temperature-reduction effect was higher than the air temperature-increase effect due to urban development.

Through scenario analyses, this study investigated air temperature variations according to urban spatial changes. The case study results suggest that to improve the thermal environment of the study area, additional urban development should be strictly controlled and green roofs should be expanded as much as possible.

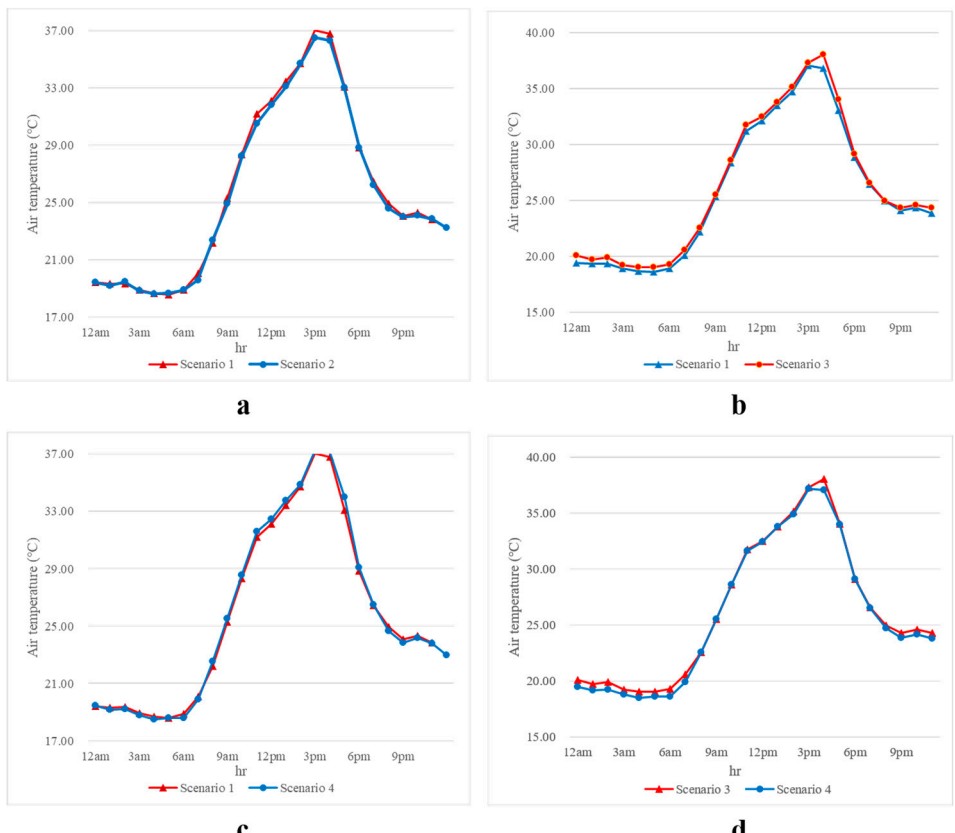

**Figure 8.** Comparison of simulated air temperature on each scenario (24 h). (**a**) Air temperature ranges of scenario 1 and scenario 2, (**b**) Air temperature ranges of scenario 1 and scenario 3, (**c**) Air temperature ranges of scenario 1 and scenario 4, (**d**) Air temperature ranges of scenario 3 and scenario 4.

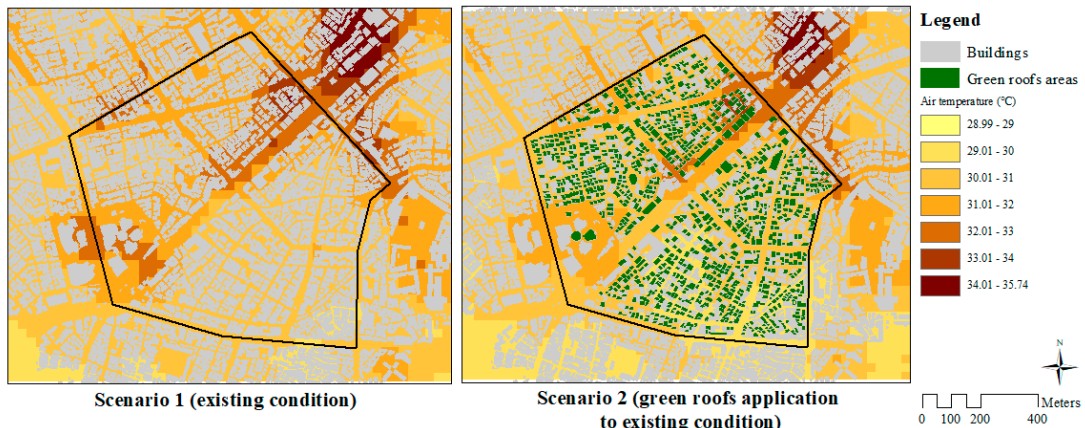

**Figure 9.** Comparison of simulated air temperature between scenario 1 (**left**) and scenario 2 (**right**) (at 11 a.m.).

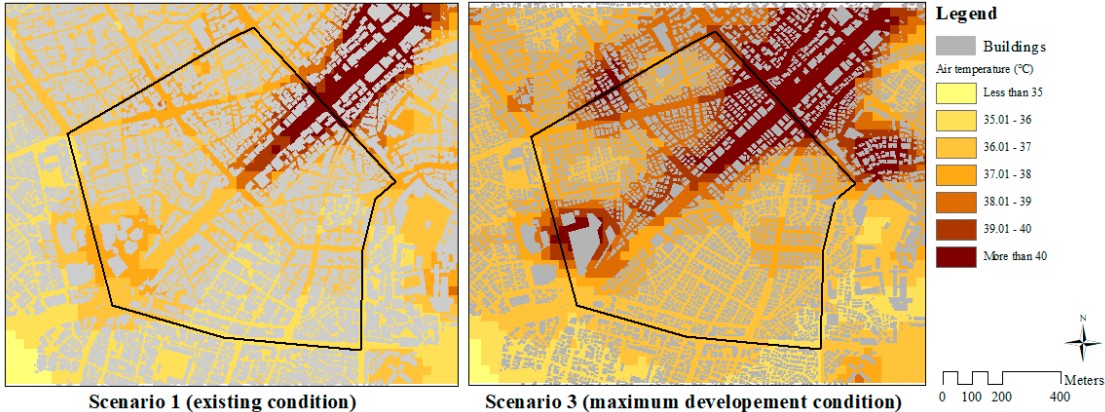

**Figure 10.** Comparison of simulated air temperature between scenario 1 (**left**) and scenario 3 (**right**) (at 16:00).

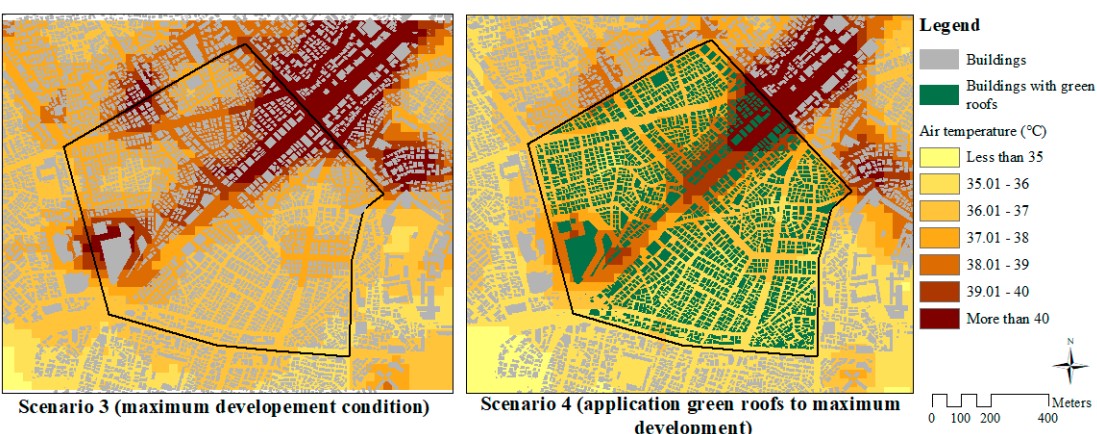

**Figure 11.** Comparison of simulated air temperature between Scenario 3 (**left**) and Scenario 4 (**right**) (at 16:00).

## 4. Discussion and Conclusions

As a result of applying the UTEMP system to the area where the heat island phenomenon occurs frequently in the city of Seoul, it was found that the air temperature simulation results were fairly similar to real observations. In addition, this study identified air temperature variations according to characteristics of urban spatial changes.

Based on the case study results, the UTEMP system was found to be useful in the following three ways: Firstly, the UTEMP system showed a simulation error of RMSE ± 0.91, so it can be sufficiently utilized for urban meteorological analysis. Previous MCSMs including the urban canopy model or meso-scale model assumed an urban area has a homogeneous array of buildings or an urban canopy layer that has even roughness. The thermal environment analysis engine of the UTEMP system considers urban structure to be a variety of morphological types on the district scale-level (between meso-scale and micro-scale), so it reflects urban spatial characteristics more concretely.

Secondly, the UTEMP system enables analysis of differentiated results of air temperature both for the change of land cover types such as rooftop greening, and the change of microscopic urban spaces such as the location and the shape of building. In order to achieve practical improvement of the thermal environment in urban spaces, introduction of thermal environment improvement measures as well as future urban spatial changes including urban development density should be considered. The results of temperature variation analyses by diverse urban spatial change scenarios suggest that it can be used as a scientific and practical planning tool that considers thermal environment improvement of urban planning and design.

Thirdly, by combining a GIS engine and thermal environment analysis engine, the UTEMP system developed in this study makes it easier for non-expert users related to the climate field to analyze urban meteorological phenomena. Unlike previously developed climate modeling tools, GIS files that are mainly used in the urban planning and design field can be directly inputted to the system without tedious pre-processing, and such an abbreviated process allows climate analysis to be more efficient. Such characteristics of this system have the advantage of instantly reflecting the changed alternatives by the urban planning and designing process to the modeling and confirms the results.

As negative environmental problems such as extreme heat events caused by urban activities and climate change intensify, the importance of scientific technology development that can effectively analyze the problem of the urban thermal environment is continuously increasing. The UTEMP developed through this study is expected to contribute to the development of MCSMs on the district scale as an initial attempt to integrate the meso- and micro-scales, which have been continuously discussed in MCSM review studies. In addition, while previous MCSMs focused on meteorological modeling, the implementation of functions for urban planning and design has been insufficient. The UTEMP is expected to contribute to the establishment of alternatives to improve the urban thermal environment by integrating the urban planning design functions and the MCSM function into one system.

On the other hand, the following three study limitations should be overcome to apply the UTEMP system to urban planning and design.

Firstly, the accuracy of the UTEMP system should be improved and multiple model time should be considered in model verifications. The analysis results presented are based on an analysis performed on a specific date. After the system has been sufficiently improved, more verification analysis of multiple dates including the winter season (rather than limited to specific days) could be possible. In that case, the error range is expected to be improved.

Secondly, the applicability of UTEMP to diverse urban spatial characteristics (e.g., different land cover, building type, road patterns, etc.) should be verified in order to secure usefulness for urban design usage. Although the study area in this study is where the heat island phenomenon occurs frequently among the representative heat island cities in Korea, there are many other areas that cause heat islands with different urban spatial characteristics. The model parameters for calculating air temperature were chosen based on previous empirical studies. In addition, LPDM (which has been applied around the world to predict wind simulation) was applied to create the wind field. As such, we expect to develop UTEMP to be applied to other cities. If the verification process is performed sufficiently for spaces composed of different land cover and diverse building heights and densities, the parameter values applied in this study could be more precisely adjusted.

Finally, it is necessary to analyze and verify the effects of urban thermal environment improvement measures. In this study, only the change of development density and the application of green roofs were considered as the urban space change scenario. In addition to rooftop greening, it is necessary to investigate the effects of other heat reduction measures (wind corridor creation, cool roofs and pavements, etc.), which have recently been attracting attention as countermeasures to reduce urban heat islands.

**Author Contributions:** This article is the result of the joint work by all authors. K.O. supervised and coordinated work on the paper. All authors conceived, designed, and carried out the methods selection and analyzed the data. All authors prepared the data visualization and contributed to the writing of this paper. All authors discussed and agreed to submit the manuscript.

**Funding:** This research was funded by the Ministry of Land, Infrastructure, and Transport of the Korean Government grant number 19AUDP-B102406-05.

**Acknowledgments:** This research was supported by a grant (19ADUP-B102560-05) from the Architecture & Urban Development Research Program (AUDP) funded by the Ministry of Land, Infrastructure and Transport of the Korean government.

**Conflicts of Interest:** The authors declare no conflict of interest.

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
