# Peer review of "Developing the Urban Thermal Environment Management and Planning (UTEMP) System to Support Urban Planning and Design†"

_sustainability, doi:10.3390/su11082224_

Round 1
Reviewer 1 Report
This paper focuses on an important topic that should be of interest to the readers. I have a few issues/recommendations that should be addressed:
Page 2 line 87: cite
Page 3 lines 92-99: Hypothesis and objective are not clearly stated.
Page 14line 343: please discuss study limitations.
References
You should include more references in your introduction, for example:
https://doi.org/10.1016/j.rser.2017.05.248
https://doi.org/10.1016/j.uclim.2018.09.001
https://doi.org/10.1016/j.solener.2018.05.015
https://doi.org/10.1016/j.buildenv.2016.11.026
https://doi.org/10.1016/j.scs.2015.04.001
Author Response
Response to Reviewer 1 Comments
Point 1: Page 2 line 87: cite
Response 1:
Related references are now cited.
Point 2: Page 3 lines 92-99: Hypothesis and objective are not clearly stated.
Response 2:
(line 93-104) The hypothesis and objective of this study are rewritten clearly.
Point 3: Page 14line 343: please discuss study limitations.
Response 3:
(line 371-392) The study limitations are discussed more concretely.
Point 4: References
You should include more references in your introduction, for example:
https://doi.org/10.1016/j.rser.2017.05.248; https://doi.org/10.1016/j.uclim.2018.09.001; https://doi.org/10.1016/j.solener.2018.05.015; https://doi.org/10.1016/j.buildenv.2016.11.026; https://doi.org/10.1016/j.scs.2015.04.001
Response 4:
(line 41- 104) The suggested references were included in the “Introduction”.

Reviewer 2 Report
I have reviewed the manuscript "Developing the Urban Thermal Environment Management and Planning (UTEMP) System to Support Urban Planning and Design". The article is a mathematical climate simulation modeling concept and is very important to investigating urban heat island phenomenon. By combining meteorological modeling and GIS analysis methods, this study developed the Urban Thermal Environment Management and Planning (UTEMP) system. The use of the UTEMP system is research that brings novelty and interest to this discipline. I would recommend this manuscript for publication after a few minor revisions.
Figure 1 : should be clearer, the image should have a higher resolution
Throughout the article you should include a space between the number and unit.
e.g., line 176 it should be (0.8 km2) and line 179 should be like this: (3.5 km × 2.5 km, 8.75 km2).
Please correct this grammatical error throughout the manuscript
other grammatical errors include: line 202 should be 06:00 am
line 214 should be e.g., 41.3 %
Table 4. should have a space between the number of percentage
line 269 should be 18.57 to 37.03 °C
LINES 344-351 is the concluding statement. the authors should expand their concluding remarks by further stating its importance and write a few sentences explaining why UTEMP is important. From the last sentence,...
"Finally, if the applicability of UTEMP is verified in both the micro and 350 macro-scale urban areas, the utilization of the system will be further increased."
Please expand upon this to better clarify its applicability in a micro and macro sense.
Author Response
Response to Reviewer 2 Comments
Point 1: Figure 1 : should be clearer, the image should have a higher resolution
Response 1:
The resolution of figure 1 has been improved.
Point 2: Throughout the article you should include a space between the number and unit.
e.g., line 176 it should be (0.8 km2) and line 179 should be like this: (3.5 km × 2.5 km, 8.75 km2).Please correct this grammatical error throughout the manuscript
other grammatical errors include: line 202 should be 06:00 am, line 214 should be e.g., 41.3 %
Table 4. should have a space between the number of percentage, line 269 should be 18.57 to 37.03 °C
Response 2:
A space between the number and unit has been added throughout the manuscript.
Point 3: LINES 344-351 is the concluding statement. the authors should expand their concluding remarks by further stating its importance and write a few sentences explaining why UTEMP is important. From the last sentence,...
"Finally, if the applicability of UTEMP is verified in both the micro and macro-scale urban areas, the utilization of the system will be further increased."
Please expand upon this to better clarify its applicability in a micro and macro sense.
Response 3:
(line 361-392) The importance of UTEMP has been more clearly discussed in “Discussions and Conclusions”. In addition, the limitations of this study based on the reviewer’s comments have also been added.

Reviewer 3 Report
This is a highly interesting piece of work. I admire the authors for developing a useful took that can inform practice. However, I cannot recommend publication of the manuscript at its current status for the following reasons.
First, I question whether the UTEMP system can be applied elsewhere other than the selected neighborhood in Seoul. Can it be applied to other parts of the city or even cities in other countries with differing climate conditions? Would the results still be reliable?
Second, somewhat related to my first comment, I question whether the selected neighborhood is highly representative of Seoul. My understanding of the city is that it is filled with high-rise apartment buildings to accommodate a population of 10 million in a dense setting. But my observation of the neighborhood does not tell that.
Third, I question why the authors chose early June for verification (or validation). Is it the hottest season of the year, or a period with modest climatic conditions? I don’t see any clear reason. In this sense, I wonder whether the UTEMP system can be applied to the coldest season. Again, would the results still be reliable?
Overall, I keep thinking that the verification (or validation) procedure presented in the manuscript is not thorough and comprehensive enough to make the system to be acceptable for implementing in state-of-the-art research or making critical decisions about local planning and design policies to combat climate change and enhance people’s outdoor comfort.
I also wonder why bigger differences occur between AWS data and simulated results especially in the afternoon in Figure 7. This part requires clarification. Otherwise, the system cannot be applied on afternoons. That’s my observation. I note that it is extremely not wise to simply mention that the errors are small and comparable to those found in other studies in line 257.
Lastly, English needs major improvement throughout the manuscript.
Author Response
Response to Reviewer 3 Comments
Point 1: First, I question whether the UTEMP system can be applied elsewhere other than the selected neighborhood in Seoul. Can it be applied to other parts of the city or even cities in other countries with differing climate conditions? Would the results still be reliable?
Response 1:
(line 378-387) As a study limitation, the UTEMP system requires further study to verify whether it can be applied to other cities. The need for applying the UTEMP system to diverse urban spatial characteristics are suggested in “Discussions and Conclusions”. In addition, the UTEMP system developed thus far is the initial version, and thus the parameters of land cover for calculating air temperature were chosen based on previous empirical studies, and LPDM (Lagrangian Particle Dispersion Model) which has been applied around world to predict wind simulation was applied to create wind field. As a result, we expect to develop MCSM (Mathematical climate simulation modelling) and subsequently be able to apply it to other cities.
Point 2: Second, somewhat related to my first comment, I question whether the selected neighborhood is highly representative of Seoul. My understanding of the city is that it is filled with high-rise apartment buildings to accommodate a population of 10 million in a dense setting. But my observation of the neighborhood does not tell that.
Response 2:
(line 179-186) While one typical urban spatial characteristic of Seoul is its high-rise apartments, the reason for selecting the study area is as follows:
Seoul does not only consist of high-rise residential apartments, but is also a city with various land uses and diverse building heights and densities. Therefore, we believe that an area which has mixed land uses and buildings heights is more appropriate to verify the usefulness of the UTEMP system. In addition, high-rise apartments are just one typical spatial type in Seoul and not the only representative type of space that causes the heat island problem of the city. Since the objective of this study is to improve the thermal environment in urban areas, the area that is representative of the heat island effect in Seoul was chosen. Still, some high-rise apartments were also included in the model area.
Point 3: Third, I question why the authors chose early June for verification (or validation). Is it the hottest season of the year, or a period with modest climatic conditions? I don’t see any clear reason. In this sense, I wonder whether the UTEMP system can be applied to the coldest season. Again, would the results still be reliable?
Response 3:
(line 212-215) In order to verify the accuracy of MCSM, it is necessary to select the day that minimizes external effects including clouds, precipitations, and wind speed. In general, the summer season in the study area is June to August, and this study selected the least cloudy days with gentle wind speed during that period.
(line 373-377) In addition, the objective of this study is to improve the thermal environment in summer, and thus, the winter season was not considered. This is mentioned as the study limitation in “Discussion and Conclusions”.
Point 4: Overall, I keep thinking that the verification (or validation) procedure presented in the manuscript is not thorough and comprehensive enough to make the system to be acceptable for implementing in state-of-the-art research or making critical decisions about local planning and design policies to combat climate change and enhance people’s outdoor comfort.
Response 4:
(line 211-226) To validate the model verifications, data from AWSs which exist and are located in different urban spatial characteristics were obtained and compared with simulated data. This has been more elaborated upon in “2.2.2 Accurate verification of the climate model results”.
Point 5: I also wonder why bigger differences occur between AWS data and simulated results especially in the afternoon in Figure 7. This part requires clarification. Otherwise, the system cannot be applied on afternoons. That’s my observation. I note that it is extremely not wise to simply mention that the errors are small and comparable to those found in other studies in line 257.
Response 5:
(line 270-278) The UTEMP system is a first version and thus accuracy verification is underway to reduce errors with the actual data. We are also aware of the increasing gap with AWS at certain times. The presented analysis results are based on analysis performed on specific dates, and the Adj. R2 are similar to the previous studies. However, as noted by the reviewer, it was observed that the air temperatures at a certain time are somewhat higher, and this phenomenon is an improvement in many studies related to MCSM.
(line 375-377) After the system has been sufficiently improved, we will conduct a more thorough verification analysis of multiple dates including winter season rather than be limited to specific days. Therefore, such an error range is expected to be improved.
Point 6: Lastly, English needs major improvement throughout the manuscript.
Response 6:
The manuscript has been edited by a native English speaker.

Round 2
Reviewer 3 Report
The very first sentence I usually put at the beginning of a response (or a 2nd-round review) to a revised manuscript is that I thank the authors for reflecting my comments and suggestions and making relevant updates. However, for this revision, it is extremely difficult for me to do so because none of my comments and suggestions seem to have been reflected. Most of what I pointed out previously was on representativeness and validation issues, and I don't see anything updated but excuses. I question to what the degree the authors are taking my comments and suggestions seriously.
No matter what the other reviewers' suggestions may be, I still do not find the manuscript acceptable for publication. There isn't much revised from the previous version. It is up to the editor to make the final decision.